

# Comparative and evolutionary analysis of the reptilian hedgehog gene family (*Shh*, *Dhh*, and *Ihh*)

Tian Xia[1], Honghai Zhang[1], Lei Zhang[1], Xiufeng Yang[1], Guolei Sun[1], Jun Chen[2], Dajie Xu[1] and Chao Zhao[1]

[1] College of Life Science, Qufu Normal University, Qufu, Shandong, China
[2] College of Marine Life Science, Ocean University of Qingdao, Qingdao, Shandong, China

## ABSTRACT

The hedgehog signaling pathway plays a vital role in human and animal patterning and cell proliferation during the developmental process. The hedgehog gene family of vertebrate species includes three genes, *Shh*, *Dhh*, and *Ihh*, which possess different functions and expression patterns. Despite the importance of hedgehog genes, genomic evidence of this gene family in reptiles is lacking. In this study, the available genomes of a number of representative reptile species were explored by utilizing adaptive evolutionary analysis methods to characterize the evolutionary patterns of the hedgehog gene family. Altogether, 33 sonic hedgehog (*Shh*), 25 desert hedgehog (*Dhh*), and 20 Indian hedgehog (*Ihh*) genes were obtained from reptiles, and six avian and five mammalian sequences were added to the analysis. The phylogenetic maximum likelihood (ML) tree of the *Shh*, *Dhh*, and *Ihh* genes revealed a similar topology, which is approximately consistent with the traditional taxonomic group. No shared positive selection site was identified by the PAML site model or the three methods in the Data Monkey Server. Branch model and Clade model C analyses revealed that the *Dhh* and *Ihh* genes experienced different evolutionary forces in reptiles and other vertebrates, while the *Shh* gene was not significantly different in terms of selection pressure. The different evolutionary rates of the *Dhh* and *Ihh* genes suggest that these genes may be potential contributors to the discrepant sperm and body development of different clades. The different adaptive evolutionary history of the *Shh*, *Dhh*, and *Ihh* genes among reptiles may be due to their different functions in regulating cellular events of development from the embryonic stages to adulthood. Overall, this study has provided meaningful information regarding the evolution of the hedgehog gene family in reptiles and a theoretical foundation for further analyses on the functional and molecular mechanisms that have shaped the reptilian hedgehog genes.

Corresponding author
Honghai Zhang,
zhanghonghai67@126.com

## INTRODUCTION

The hedgehog signaling pathway is described as "one of the most enigmatic" pathways, although we know that it is essential for patterning and cell proliferation in humans and animals (*Hooper & Scott, 2005*). Hedgehog genes were initially identified during genetic

screenings of *Drosophila melanogaster*, with only a small number of genes in this family being isolated (*Nüsslein-Volhard & Wieschaus, 1980*). Hedgehog orthologs from vertebrates, such as *Mus musculus* (mouse) and *Danio rerio* (zebrafish), were cloned in 1993 (*Echelard et al., 1993*; *Krauss, Concordet & Ingham, 1993*). Duplication of the vertebrate genome contributed to the expansion of the hedgehog gene family, which can be classified into three subgroups: the sonic hedgehog (*Shh*), Indian hedgehog (*Ihh*), and desert hedgehog (*Dhh*) genes (*Kumar, Balczarek & Lai, 1996*).

The different functions of these hedgehog genes result mainly from their discrepant expression patterns (*Varjosalo & Taipale, 2008*). Namely, the *Shh* gene plays a core role in the development and patterning of the skeletal and nervous systems (*Ingham & McMahon, 2001*), and all systems have undergone remarkable morphological changes in primates, especially in humans (*Dorus et al., 2006*). The *Ihh* gene regulates the formation of vascular and endochondral ossification, and *Dhh* is indispensable for the formation of the peripheral nervous system (*Nagase et al., 2010*), as well as the differentiation of peritubular myoid cells and the formation of the embryonic testis cord (*Humphrey Hung-Chang, Wendy & Blanche, 2002*). Vertebrate hedgehog genes are expressed in different tissues during the developmental process. Sonic hedgehog gene expression is highest in the growing organs of the foregut, including the lungs, pancreas, esophagus, liver, and proximal stomach (*Litingtung et al., 1998*; *Motoyama et al., 1998*). Indian hedgehog gene expression occurs in hindgut-induced tissues, especially the distal stomach, coelom, and intestines (*Zacharias et al., 2010*). The *Dhh* gene is crucial for the maintenance of male spermatogenesis and the germline (*Szczepny, Hime & Loveland, 2010*) and is also expressed in Schwann cells, taking part in the formation of the nerve sheath (*Mirsky et al., 2010*). Each gene possesses diverse expression patterns and a specific biological function, making them play diverse roles in different vertebrate taxa (*Shimeld, 1999*).

Previous research on the molecular evolution of the hedgehog gene family in invertebrates revealed positive selection signals, which appeared to be associated with the divergence of the two major bilaterian groups, Deuterostomia and Ecdysozoa (*Gunbin, Afonnikov & Kolchanov, 2007*). Comparative genomics was used to characterize the evolution of the hedgehog genes in 45 avian and three reptilian genomes, suggesting that hedgehog paralogous genes in vertebrates evolved independently within homologous linkage groups at different evolutionary rates (*Pereira et al., 2014*). Previous research has focused on the functional divergence of the hedgehog pathway during evolution and its integration with other signaling pathways to control and regulate cell growth, differentiation, and survival (*Beachy, Karhadkar & Berman, 2004*; *Ingham & McMahon, 2001*). Recently, the function of the hedgehog pathway became a research hotspot, which resulted in an accumulation of knowledge regarding human disorders, including cancer and birth defects (*Mcmahon, Ingham & Tabin, 2003*; *Nieuwenhuis & Hui, 2005*). Moreover, research on the hedgehog gene family has been extensively studied in cnidaria (*Matus et al., 2008*), zebrafish (*Avaron et al., 2006*), amphioxi (*Shimeld, 1999*), and avians (*Pereira et al., 2014*).

Reptiles have a unique physiological feature as the sole poikilothermic amniotes (*Zimmerman, Vogel & Bowden, 2010*). They play essential roles in their ecosystem as prey,

predators, grazers, and commensal species, serving as the perfect model to investigate the biological and evolutionary histories underlying speciation. Despite the prominent role of reptiles in the lengthy evolutionary history of vertebrates, the reptilian hedgehog signaling pathway has received little attention. Previously, two reptiles, the American alligator and green turtle, were used in the evolutionary analysis of the avian hedgehog gene family (*Pereira et al., 2014*). With the rapid development of high-throughput sequencing, the available genomes of a number of reptiles have been published, which provides a new opportunity for the investigation of evolutionary patterns and structural characteristics of reptilian hedgehog genes. Given the current lack of comprehensive studies on the hedgehog gene family molecular evolution across the reptilian phylogeny, here we performed a comparative evolutionary analysis on available reptilian whole-genome sequences representing three reptilian orders. Our detection of natural selection acting on three members of the hedgehog gene family in reptiles unraveled adaptive evolution at the molecular level.

## MATERIALS AND METHODS

### Species and sequences

The *Shh*, *Ihh*, and *Dhh* gene sequences were retrieved from the National Center for Biotechnology Information database (www.ncbi.nlm.nih.gov). Meanwhile, the *Shh*, *Ihh*, and *Dhh* amino acid sequences of multiple species, including *Pelodiscus sinensis* (Chinese soft-shelled turtle), *Pseudonaja textilis* (eastern brown snake), *Notechis scutatus* (mainland tiger snake), *Terrapene Mexicana* (Mexican box turtle), *Pogona vitticeps* (central bearded dragon), *Protobothrops mucrosquamatus* (Taiwan habu), *Anolis carolinensis* (green anole), *Alligator mississippiensis* (American alligator), and *Chrysemys picta* (painted turtle) were treated as queries to detect the genome sequences of *Crocodylus porosus* (Australian saltwater crocodile), *Malaclemys terrapin* (diamondback terrapin), *Cuora mccordi* (McCord's box turtle), *Chelonoidis abingdonii* (Abingdon island giant tortoise), *Gopherus agassizii* (Agassiz's desert tortoise), *Platysternon megacephalum* (big-headed turtle), *Crotalus viridis* (western rattlesnake), *Vipera berus* (European adder), *Salvator merianae* (Argentine black and white tegu), *Lacerta viridis* (green lizard), *Protobothrops flavoviridis* (habu), *L. bilineata* (western green lizard), *Paroedura picta* (panther gecko), *Pantherophis guttatus* (corn snake), *Ophiophagus hannah* (king cobra), *Crotalus pyrrhus* (white speckled rattlesnake), *Hydrophis cyanocinctus* (Asian annulated sea snake), *H. hardwickii* (Hardwick's sea snake), *Thermophis baileyi* (Bailey's Snake), and *Ophisaurus gracilis* (Anguidae lizard) (*Gilbert et al., 2014*; *Green et al., 2014*; *Hara et al., 2018*; *Li et al., 2018*; *Pasquesi, Adams & Card, 2018*; *Roscito et al., 2018*; *Shibata et al., 2018*; *Song et al., 2015*; *Tollis et al., 2017*; *Ullate-Agote, Milinkovitch & Tzika, 2014*; *Vonk et al., 2013*). Reptiles were divided into three disparate orders, including Crocodylia, Testudines, and Squamata. The available information on reptilian genomes was integrated (Table S1). The commands BLASTN and TBLASTN in BLAST v2.7.1 were used for orthologous gene searches with an E-value of $1e^{-10}$ in the aforementioned reptilian genomes. Identification of the hedgehog genes was determined by a previously established method (*Monie, Gay & Gangloff, 2008*). Additionally, the hedgehog gene sequences of birds and other

vertebrates were obtained for further analysis. Information regarding accession numbers and sequences of hedgehog genes is provided in Table S2.

## Phylogenetic analysis

Multiple sequence alignment of the three hedgehog genes was analyzed by the Multiple Sequence Comparison by Log-Expectation (www.ebi.ac.uk/Tools/msa/muscle). Phylogenetic trees were constructed to estimate the differentiation among the three hedgehog genes, using RAxML v8.2.12 with 1,000 bootstrap replications and a GTR+Γ sequence evolution model. Relevant phylogenetic relationships among reptiles were obtained using Timetree v3.0 (www.timetree.org). Moreover, a comparison of the *Shh*, *Dhh*, and *Ihh* gene trees with the reptilian species tree was conducted.

## Test for positive selection

Based on the corresponding topology of the aforementioned species, the phylogenetic analysis by maximum likelihood (PAML) in Codeml v4.9d with codon-based likelihood models was used to estimate nonsynonymous/synonymous substitution ratios ($\omega$ = dN/dS) to quantify natural selection for the hedgehog genes. Different $\omega$ values represent different types of selection, namely, $\omega < 1$ represents purifying selection, $\omega = 1$ represents neutral selection, and $\omega > 1$ represents positive selection (*Kimura, 1979*). $P < 0.05$ was applied to estimate whether the alternative selective hypothesis was significant or not. Previous studies have focused on the function of hedgehog genes during the developmental process, mainly in the nervous system, bones, skin, and pancreas (*Ingham, Nakano & Seger, 2011*; *Varjosalo & Taipale, 2008*). The hedgehog proteins control the growth, survival, and fate of cell, and pattern of vertebrate body plan (*Varjosalo & Taipale, 2008*). Reptiles have different physiological characteristics, which indicate that there are different types of development. In order to determine branch-specific evolutionary rates, we performed the branch model (two-ratio vs. one-ratio model) in CODEML program to evaluate $\omega$ ratio for each branch. In addition, to detect the probabilities of sites under positive selection in each linage, branch-site model was used in which the $\omega$ ratio could vary among sites in divergent clades of reptilian species. To further contrast the evolutionary rates of hedgehog genes in response to divergent clades, we implemented Clade Model C (CmC), which allows different evolution along the phylogeny. Hence, we partitioned all species according to the appearance of the limb: limbless species and species with limbs. The best-suited model allowing discrepant evolutionary rates was compared with the null model M2a_rel, which has an unconstrained $\omega$.

Additionally, three evolutionary models (M7, M8a, and M8) in PAML were used to estimate site-specific selection. The likelihood ratio test (LRT) results of two nested models were compared to detect dramatic events of positive selection with two degrees of freedom (*Yang et al., 2000*). Amino acids with selection pressures were detected using a Bayes empirical Bayes approach by counting the posterior probability with site model M8 (*Ziheng, Wong & Nielsen, 2005*). Moreover, the Data Monkey Server (www.datamonkey.org) was used to test for positive selection in reptilian hedgehog genes as a supplemental method. Finally, hedgehog genes were analyzed through three distinct methods in the

HyPhy package of the Data Monkey Server, including the fixed-effect likelihood (FEL), random effect likelihood (REL), and single likelihood ancestor counting (SLAC) methods, which were used to detect positive selection (*Pond & Frost, 2005*). Default settings for significance levels of $p = 0.1$ for SLAC and FEL, and Bayes factor > 50 for REL, were used for screening positive selection sites. Sites estimated to be under positive selection were determined by at least two of the four methods.

## RESULTS

### Hedgehog gene sequences

The hedgehog genes of *Crocodylus porosus*, *Chelonoidis abingdonii*, *Crotalus viridis*, *Cuora mccordi*, *G. agassizii*, *L. bilineata*, *L. viridis*, *M. terrapin*, *Pantherophis guttatus*, *Paroedura picta*, *Platysternon megacephalum*, *Protobothrops flavoviridis*, *S. merianae*, *Ophiophagus hannah*, *Crotalus pyrrhus*, *H. cyanocinctus*, *H. hardwickii*, *T. baileyi*, *Ophisaurus gracilis*, and *V. berus* have not been previously reported. No intact *Dhh* gene was identified in Crocodylia, and no intact *Ihh* gene was identified in Crocodylia and Testudines. After all partial genes were removed, 33 *Shh*, 20 *Ihh*, and 25 *Dhh* intact genes were obtained (Table S2). Additionally, six avian and five vertebrate sequences of each hedgehog gene were obtained with high quality. The percent identity matrix ranged from 67.3% to 100.0% for the *Shh* genes, 77.9–99.8% for the *Ihh* genes, and 70.6–99.8% for the *Dhh* genes (Table S3). The similarity between the newly obtained sequences and the previously obtained query sequences ranged from 67.3% to 100.0%, suggesting that the newly identified sequences of the three hedgehog genes were credible.

### Phylogenetic analysis

A total of 44 *Shh*, 36 *Dhh*, and 31 *Ihh* genes were obtained for building the phylogenetic tree. The maximum likelihood tree of the 111 hedgehog genes revealed that the 44 *Shh* genes were divided into six primary clades, Serpentes, Sauria, Crocodylia, Testudines, Aves, and Mammalia (Fig. 1), while the *Dhh* genes were divided into four major clades, Squamata, Testudines, Aves, and Mammalia (Fig. S1), and the *Ihh* genes were divided into four different clades, Serpentes, Sauria, Aves, and Mammalia (Fig. S2). The phylogenetic tree of reptiles, birds, and other vertebrates downloaded from Timetree divided reptiles into four clades according to their catalogs. Then, the hedgehog gene tree was compared to the species trees, revealing that the phylogenetic proximity of the *Shh*, *Dhh*, and *Ihh* genes tended to be consistent with the traditional taxonomic group.

### Identification of selection pressure for *Shh*, *Dhh*, and *Ihh*

The site model analysis did not detect positive selection in the three hedgehog genes (Table S4). The best nucleotide substitution bias models of the three hedgehog genes were then executed in Data Monkey Server with three methods. Again, no positive selection site was found by the FEL and SLAC methods with a significance level of 0.1, which supports the M8 model findings. However, the REL method did identify six sites in the *Dhh* gene, and the SLAC method detected one site, but neither of these methods could confirm identical sites.
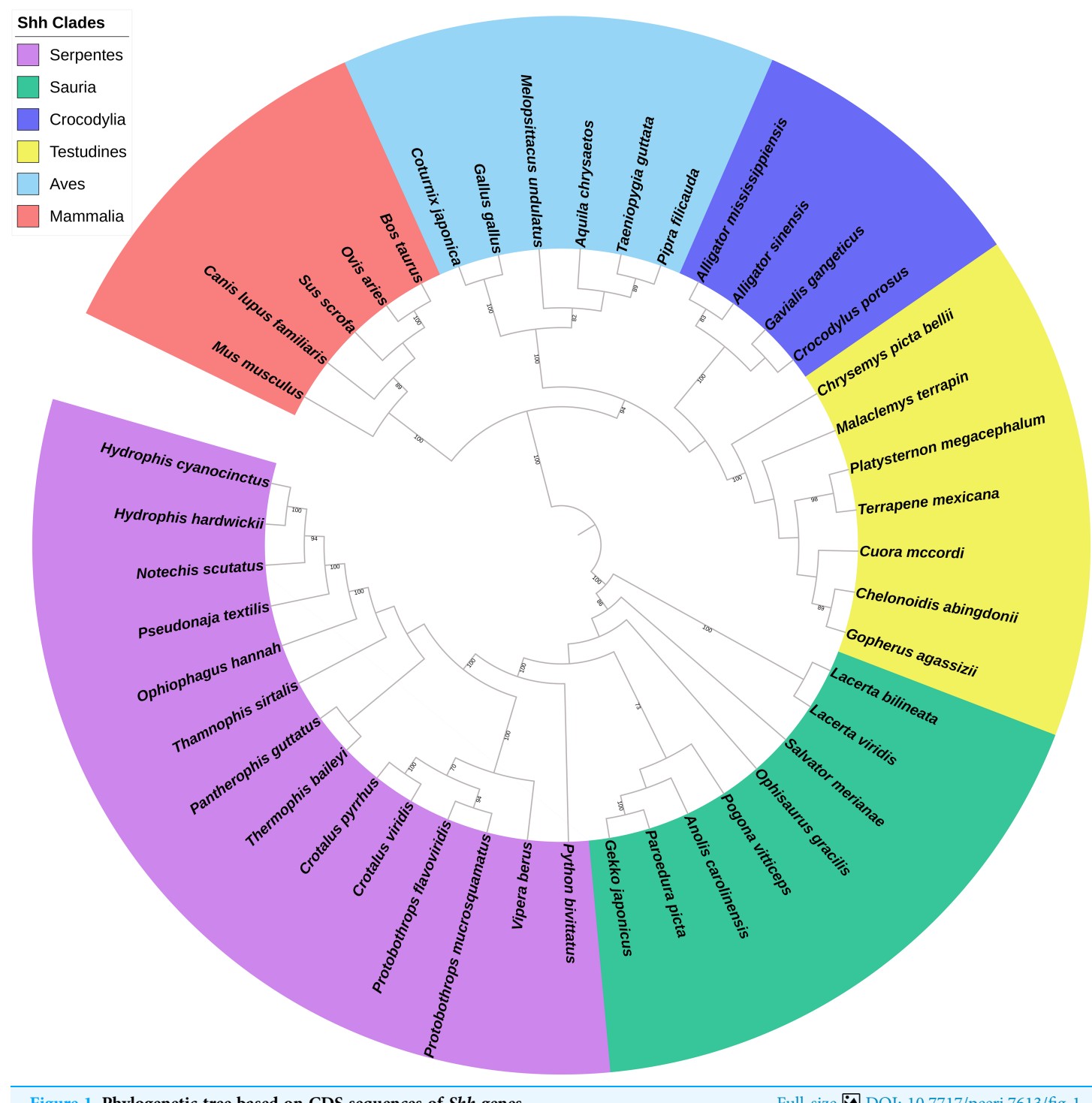

**Figure 1  Phylogenetic tree based on CDS sequences of *Shh* genes.**

The branch model was next used to evaluate the evolutionary rates of each specific branch; the targeted branches were assigned as foreground and the others were set as background branches. We set different orders as independent lineages, according to the topology of reptiles. The branch model analysis revealed that *Dhh and Ihh* had significantly

**Table 1  Test for positive selection in divergent clades of *Shh*, *Dhh*, and *Ihh* genes with branch model.**

| Gene | Model compared | np | LnL | LRT *p*-value | ω for branch |
|------|----------------|----|-----|---------------|--------------|
| *Shh* | M0 | 88 | −11647.017474 | | $\omega_1 = 0.07001$ |
| | M2(clade of Crocodylia) | 89 | −11646.585976 | 0.352900523 | $\omega_1 = 0.05660$; $\omega_2 = 0.07081$ |
| | M2(clade of Testudines) | 89 | −11646.736015 | 0.453094143 | $\omega_1 = 0.09345$; $\omega_2 = 0.06965$ |
| | M2(clade of Sauria) | 89 | −11645.645632 | 0.097638458 | $\omega_1 = 0.06189$; $\omega_2 = 0.07429$ |
| | M2(clade of Serpentes) | 89 | −11646.656047 | 0.395226992 | $\omega_1 = 0.07728$; $\omega_2 = 0.06869$ |
| | M2(Limbless Squamates) | 89 | −11646.431808 | 0.279134287 | $\omega_1 = 0.07844$; $\omega_2 = 0.06817$ |
| *Dhh* | M0 | 72 | −9913.616846 | | $\omega_1 = 0.07078$ |
| | M2(clade of Serpentes) | 73 | −9925.890310 | 0.000000725[a] | $\omega_1 = 0.10761$; $\omega_2 = 0.06381$ |
| | M2(clade of Sauria) | 73 | −9924.330432 | 0.000003675[a] | $\omega_1 = 0.10124$; $\omega_2 = 0.06045$ |
| | M2(Limbless Squamates) | 73 | −9926.514254 | 0.00000038[a] | $\omega_1 = 0.10077$; $\omega_2 = 0.06309$ |
| | M2(Testudines) | 73 | −9928.261444 | 0.000000062[a] | $\omega_1 = 0.11848$; $\omega_2 = 0.06720$ |
| *Ihh* | M0 | 62 | −8482.453248 | | $\omega_1 = 0.08385$ |
| | M2(clade of Serpentes) | 63 | −8475.468795 | 0.00018586[a] | $\omega_1 = 0.13019$; $\omega_2 = 0.07412$ |
| | M2(Sauria) | 63 | −8471.679842 | 0.000003453[a] | $\omega_1 = 0.13072$; $\omega_2 = 0.06886$ |
| | M2(Limbless Squamates) | 63 | −8470.418614 | 0.000000929[a] | $\omega_1 = 0.13521$; $\omega_2 = 0.06857$ |

Notes:
[a] Significant level.
Limbless Squamates mean Serpentes and *Ophisaurus gracilis*, representing the limbless clades.

different ω values among the diverse lineages of reptiles, indicating heterogeneous selective pressures on different lineages, while *Shh* had no obvious difference among lineages (Table 1). The branch-site model analysis found positive selection sites for the three hedgehog genes, but none of the lineages of *Dhh* and *Ihh* genes reached the significant level. Meanwhile, the analysis revealed that the *Shh* gene of Testudines underwent positive selection and the sites were significantly different ($p = 0.018$) (Table 2). Furthermore, CmC detected that the partitions of *Dhh* and *Ihh* genes were markedly better suited relative to the M2a_rel model ($p < 0.01$), which supports the findings of different ω rates among the partition of limbs. Comparatively, the LRTs between these two models for all *Shh* gene clades were not significant ($p = 0.074$, Table 3).

# DISCUSSION

Reptiles are a group of ancient vertebrates that embody a momentous position in the long and complex evolutionary history of vertebrates. As is widely known, reptiles cannot maintain a constant body temperature and experience seasonal shifts in behavior that are correlated with the ambient temperature (*Zimmerman, Vogel & Bowden, 2010*). There is also a complex relationship between ontogenesis and animal body temperature. In this study, the molecular evolution of the *Shh*, *Dhh*, and *Ihh* genes in reptiles was investigated. Collectively, three hedgehog gene sequences from 20 reptilian genomes were retrieved, although there are likely to be more. These genes were identified in *Crocodylus porosus*, *Chelonoidis abingdonii*, *Crotalus viridis*, *Cuora mccordi*, *G. agassizii*, *L. bilineata*, *L. viridis*, *M. terrapin*, *Pantherophis guttatus*, Paroedura *picta*, *Platysternon megacephalum*, *Protobothrops flavoviridis*, *S. merianae*, *Ophiophagus hannah*, *Crotalus pyrrhus*,

**Table 2 Test for positive selection in divergent clades of *Shh*, *Dhh*, and *Ihh* genes with branch-site model.**

| Gene | Lineage | Models compared | np | LnL | *p*-value | ω values | Positively selected sites (BEB analysis) |
|---|---|---|---|---|---|---|---|
| *Shh* | Serpentes | Model A | 91 | −11462.09307 | 1 | $p_0 = 0.03960$; $\omega_1 = 1.000$; $\omega_2 = 1.000$ | 85 I 0.970 86 F 0.524 87 K 0.980 95 D 0.907 96 R 0.948 181 I 0.938 186 K 0.842 187 A 0.931 371 L 0.567 |
| | | Model A null | 90 | −11462.09307 | | $p_0 = 0.03960$; $\omega_1 = 1.000$; $\omega_2 = 1.000$ | |
| | Sauria | Model A | 91 | −11460.7336 | 1 | $p_0 = 0.04246$; $\omega_1 = 1.000$; $\omega_2 = 1.000$ | 4 L 0.990 9 L 0.989 20 A 0.706 136 E 0.531 204 W 0.882 232 Q 0.998 240 A 0.600 244 R 0.523 301 Q 0.936 308 P 0.620 310 A 0.630 361 A 0.959 |
| | | Model A null | 90 | −11460.7336 | | $p_0 = 0.04246$ $\omega_1 = 1.000$; $\omega_2 = 1.000$ | |
| | Crocodilian | Model A | 91 | −11479.21629 | 0.511596661 | $\omega_0 = 0.04535$; $\omega_1 = 1.000$; $\omega_2 = 3.93515$ | 206 N 0.847 |
| | | Model A null | 90 | −11479.43171 | | $p_0 = 0.04546$; $\omega_1 = 1.000$; $\omega_2 = 1.000$ | |
| | Testudines | Model A | 91 | −11475.91887 | 0.017704046[a] | $p_0 = 0.04593$; $\omega_1 = 1.000$; $\omega_2 = 11.54603$ | 2 L 0.641 14 G 0.576 277 H 0.974 313 S 0.601 358 F 0.602 410 S 0.616 |
| | | Model A null | 90 | −11478.73149 | | $p_0 = 0.04569$; $\omega_1 = 1.000$; $\omega_2 = 1.000$ | |
| | Limbless Squamates | Model A | 91 | −11463.08743 | 1 | $p_0 = 0.03868$; $\omega_1 = 1.000$; $\omega_2 = 1.000$ | 85 I 0.833 87 K 0.709 96 R 0.682 186 K 0.658 187 A 0.925 244 R 0.895 258 Q 0.795 261 Q 0.524 295 H 0.828 360 I 0.780 370 L 0.995 |
| | | Model A null | 90 | −11463.08743 | | $p_0 = 0.03868$; $\omega_1 = 1.000$; $\omega_2 = 1.000$ | |
| *Dhh* | Serpentes | Model A | 75 | −9889.30068 | 1 | $p_0 = 0.05652$; $\omega_1 = 1.000$; $\omega_2 = 1.000$ | 67 V 0.965 200 R 0.741 224 Q 0.960 240 K 0.540 243 L 0.958 270 R 0.987 275 M 0.999 286 C 0.901 346 Y 0.902 |
| | | Model A null | 74 | −9889.30068 | | $p_0 = 0.05652$; $\omega_1 = 1.000$; $\omega_2 = 1.000$ | |
| | Sauria | Model A | 75 | −9871.673179 | 1 | $p_0 = 0.05115$; $\omega_1 = 1.000$; $\omega_2 = 1.000$ | 1 L 0.991 3 L 0.773 7 C 0.916 10 P 0.822 27 S 0.928 33 P 0.508 62 S 0.858 63 E 1.000 64 R 0.942 186 R 0.835 196 T 0.992 201 S 0.625 204 K 0.932 206 S 0.934 212 H 0.521 226 A 0.908 253 V 0.951 280 V 0.984 309 Q 0.744 313 T 0.982 364 A 0.802 369 V 0.658 374 Q 0.963 382 D 0.692 |
| | | Model A null | 74 | −9871.673179 | | $p_0 = 0.05115$; $\omega_1 = 1.000$; $\omega_2 = 1.000$ | |

| Gene | Lineage | Models compared | np | LnL | p-value | ω values | Positively selected sites (BEB analysis) |
|---|---|---|---|---|---|---|---|
| | Limbless Squamates | Model A | 75 | −9885.126495 | 1 | $p_0 = 0.05492$; $\omega_1 = 1.000$; $\omega_2 = 1.000$ | 2 T 0.943 67 V 0.926 112 V 0.805 200 R 0.982 209 K 0.889 224 Q 0.924 226 A 0.593 243 L 0.896 270 R 0.970 275 M 0.999 286 C 0.860 313 T 0.578 346 Y 0.819 369 V 0.531 |
| | | Model A null | 74 | −9885.126495 | | $p_0 = 0.05492$; $\omega_1 = 1.000$; $\omega_2 = 1.000$ | |
| | Testudines | Model A | 75 | −9897.057753 | 1 | $p_0 = 0.05873$; $\omega_1 = 1.000$; $\omega_2 = 1.000$ | 21 V 0.917 25 Q 0.569 35 Q 0.915 57 K 0.503 91 T 0.923 97 R 0.520 242 T 0.884 276 D 0.993 296 D 0.693 356 G 0.918 |
| | | Model A null | 74 | −9897.057753 | | $p_0 = 0.05873$; $\omega_1 = 1.000$; $\omega_2 = 1.000$ | |
| *Ihh* | Serpentes | Model A | 65 | −8316.257473 | 1 | $p_0 = 0.04207$; $\omega_1 = 1.000$; $\omega_2 = 1.000$ | 13 L 0.680 182 T 0.528 206 A 0.627 232 V 0.996 261 T 0.638 331 G 0.794 338 H 0.975 342 R 0.707 364 F 0.581 |
| | | Model A null | 64 | −8316.257473 | | $p_0 = 0.04207$; $\omega_1 = 1.000$; $\omega_2 = 1.000$ | |
| | Sauria | Model A | 65 | −8303.182547 | 1 | $p_0 = 0.03904$; $\omega_1 = 1.000$; $\omega_2 = 1.000$ | 2 E 0.983 3 V 0.590 6 C 0.736 9 S 0.607 11 R 0.639 217 A 0.702 223 V 0.598 226 R 0.987 235 Q 0.577 242 A 0.554 257 A 0.883 259 M 0.795 262 I 0.840 268 Q 0.989 271 Q 0.977 281 L 0.858 282 Q 0.920 283 P 0.837 307 L 0.741 336 Y 0.865 337 H 0.616344 W 0.991 |
| | | Model A null | 64 | −8303.182547 | | $p_0 = 0.03904$; $\omega_1 = 1.000$; $\omega_2 = 1.000$ | |
| | Limbless Squamates | Model A | 65 | −8310.549535 | 1 | $p_0 = 0.03920$; $\omega_1 = 1.000$; $\omega_2 = 1.000$ | 6 C 0.679 9 S 0.541 11 R 0.544 13 L 0.982 206 A 0.542 223 V 0.716 227 T 0.629 232 V 0.994 261 T 0.509 283 P 0.629 331 G 0.728 338 H 0.954 342 R 0.506 |
| | | Model A null | 64 | −8310.549535 | | $p_0 = 0.03920$; $\omega_1 = 1.000$; $\omega_2 = 1.000$ | |

**Notes:**
[a] Significant level.
Limbless Squamates mean Serpentes and *Ophisaurus gracilis*, representing the limbless clades.

**Table 3 Test for positive selection in divergent clades of *Shh*, *Dhh*, and *Ihh* genes with CmC model.**

| Gene | Model | np | LnL | k | Site class 0 (all branched) | Site class 1 (all branches) | Site class 2 (background branches and different clades vary) | df | *p*-value |
|---|---|---|---|---|---|---|---|---|---|
| *Shh* | M2a_rel (null) | 91 | −11291.52935 | 2.53263 | $p_0 = 0.73101$ | $p_1 = 0.01911$ | $p_2 = 0.24989$ | | |
| | | | | | $\omega_0 = 0.01124$ | $\omega_1 = 1.00000$ | $\omega_2 = 0.23489$ | | |
| | CmC | 93 | −11288.927085 | 2.53001 | $p_0 = 0.73326$ | $p_1 = 0.01964$ | $p_2 = 0.24711$ | 2 | 0.074106651 |
| | | | | | $\omega_0 = 0.01159$ | $\omega_1 = 1.00000$ | $\omega$Clade 0 (backgroud) = 0.34939 | | |
| | | | | | | | $\omega$Clade 1 = 0.24768 | | |
| | | | | | | | $\omega$Clade 2 = 0.21839 | | |
| *Dhh* | M2a_rel (null) | 75 | −9740.01743 | 3.64268 | $p_0 = 0.62809$ | $p_1 = 0.00000$ | $p_2 = 0.37191$ | | |
| | | | | | $\omega_0 = 0.00910$ | $\omega_1 = 1.00000$ | $\omega_2 = 0.19189$ | | |
| | CmC | 77 | −9719.900358 | 3.71887 | $p_0 = 0.62023$ | $p_1 = 0.00000$ | $p_2 = 0.37977$ | 2 | 0.000000002[a] |
| | | | | | $\omega_0 = 0.00795$ | $\omega_1 = 1.00000$ | $\omega$Clade 0 (background) = 0.13580 | | |
| | | | | | | | $\omega$Clade 1 = 0.25351 | | |
| | | | | | | | $\omega$Clade 2 = 0.17494 | | |
| *Ihh* | M2a_rel (null) | 65 | −8252.834363 | 3.53731 | $p_0 = 0.71133$ | $p_1 = 0.03695$ | $p_2 = 0.25172$ | | |
| | | | | | $\omega_0 = 0.01260$ | $\omega_1 = 1.00000$ | $\omega_2 = 0.25547$ | | |
| | CmC | 67 | −8247.531941 | 3.57165 | $p_0 = 0.70435$ | $p_1 = 0.03482$ | $p_2 = 0.26083$ | 2 | 0.004979628[a] |
| | | | | | $\omega_0 = 0.01177$ | $\omega_1 = 1.00000$ | $\omega$Clade 0 (background) = 0.24841 | | |
| | | | | | | | $\omega$Clade 1 = 0.37843 | | |
| | | | | | | | $\omega$Clade 2 = 0.20881 | | |

**Note:**
[a] Means significant level. Clade 1 means limbless Squamates clade, clade 2 means limbs clade.

*H. cyanocinctus*, *H. hardwickii*, *T. baileyi*, *Op*hisaurus *gracilis*, and *V. berus* for the first time. Combined with formerly predicted hedgehog genes from other reptiles, birds, and mammals, these data have provided a new perspective on the reptilian intercellular signaling pathway at the molecular level. Although the genome sequencing data have been growing and its quality is getting better, it is unavoidable that some genes will not be successfully identified due to the complexity of the gene sequence. Unfortunately, we failed to identify intact *Dhh* in Crocodilian and intact *Ihh* sequence in Crocodilian and Testudines. We believe that some of the genes may have been lost but this hypothesis remains to be established. Moreover, the homology analysis revealed that the similarity between the reptilian *Shh*, *Dhh*, and *Ihh* gene sequences ranged from 67.3% to 100.0%, which indicates that the sequences were species-specific among the subclasses of reptiles.

The phylogenetic relationship among the *Shh*, *Dhh*, and *Ihh* genes was also explored, and the results revealed that these genes could be divided into six, four, and four clades, respectively. No gene duplication was detected for these genes. When comparing the hedgehog gene trees with the species tree, the topology of the hedgehog genes tended to be consistent with the traditional taxonomic group. Thus, we speculate that the hedgehog genes were highly conserved due to their critical importance in development and that the

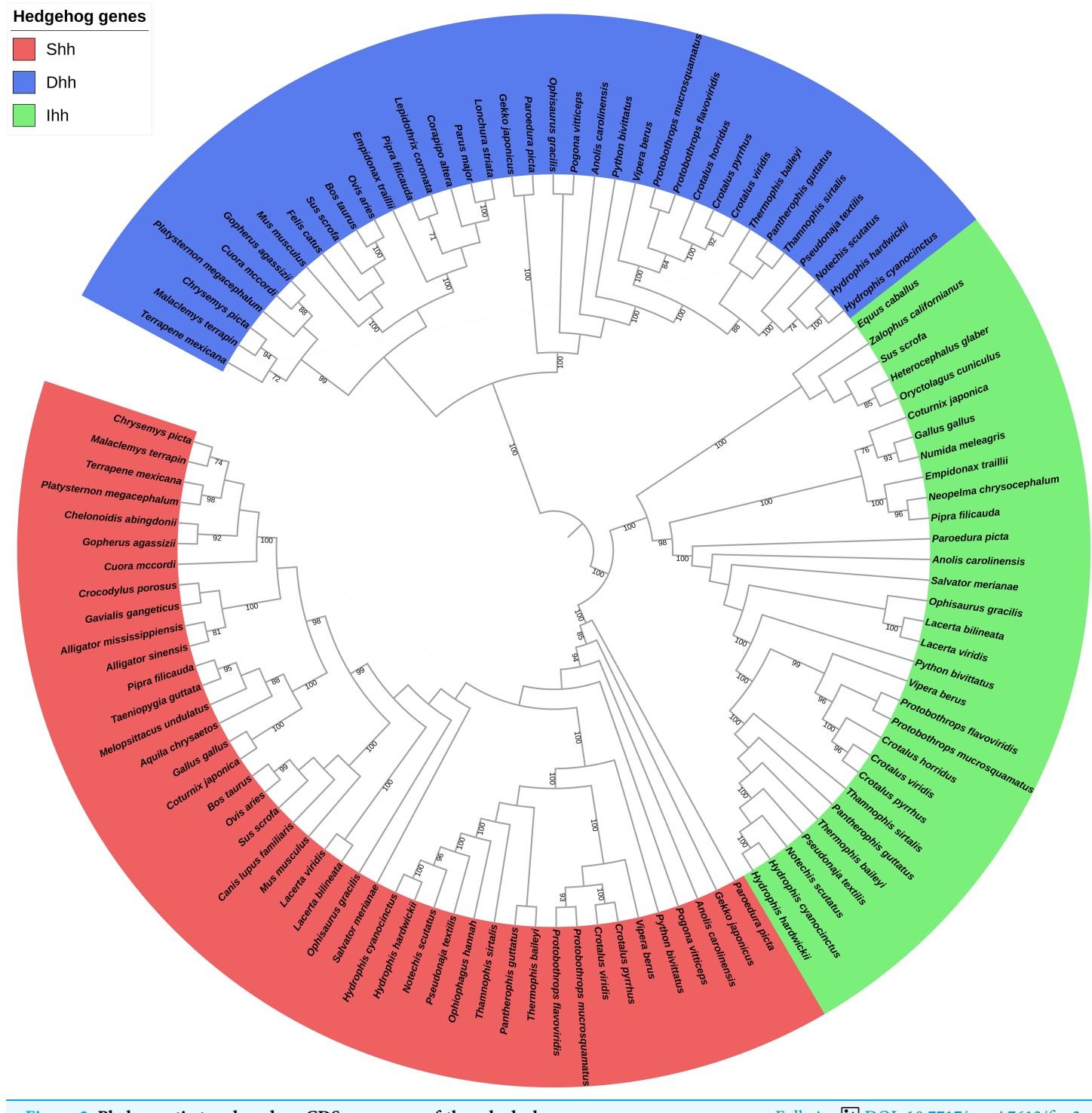

**Figure 2 Phylogenetic tree based on CDS sequences of three hedgehog genes.**

driving force behind the evolution of these genes may contribute to the differentiation of reptilian species. Additionally, the phylogenetic analysis of the three hedgehog genes' coding sequences found an overall (*Dhh*, (*Shh*, *Ihh*)) topology (Fig. 2) that is consistent

with the evolutionary history of the three hedgehog genes, as *Shh* and *Ihh* are considered to be more closely related to each other than to the *Dhh* (*Varjosalo & Taipale, 2008*).

Site models performed in PAML found that the *Shh*, *Dhh*, and *Ihh* sequences were under diverse selective pressures, with no positively selected residues. Moreover, the data did not find evidence for positive selection, despite the use of three methods implemented in the Data Monkey server to verify positive selection. As expected, no positively selected site was detected in any of the three hedgehog genes in reptiles by at least two methods. Although these methods failed to recognize residues under positive selection, several residues under negative selection were found. These results are consistent with the findings of previous studies exploring the adaptive evolution of the hedgehog gene family in vertebrates (*Pereira et al., 2014*). The purifying selection plays an essential role in the long-term stability of biological structures by deleting harmful mutations (*Karlsson, Kwiatkowski & Sabeti, 2014*). Thus, we speculate that the highly conserved characteristic of the hedgehog genes may be due to their crucial role in controlling multiple different developmental processes.

Squamate reptiles, such as snakes and snake-like lizards, are the most extreme reptilian species in terms of shape, given their elongated and limbless bodies (*Gans, 1975*). Thus, for the branch model, another branch was added in addition to the traditional classification, which included the Serpentes species and limbless *Ophisaurus gracilis*. The LRT between the null and alternate likelihood models indicated that all clades of *Dhh* and *Ihh* genes fit the two-ratio model better, implying that all lineages of these two genes may have evolved at different rates. Sonic hedgehog plays a crucial role in regulating vertebrate organogenesis, such as the development of digits on limbs. A different evolutionary rate of *Shh* was not found, as expected, which is likely because the mechanism of limb reduction in reptiles is regulated by other related genes, such as *HoxA13* (*Singarete et al., 2015*). The overall branches of *Shh* rejected the two-ratio model, suggesting that the reptilian *Shh* gene evolved at a relatively stable rate. However, six positive selection sites were detected in Testudines using the branch-site model, suggesting that this *Shh* gene underwent positive selection in Testudines. Sonic hedgehog expression in snake embryos indicates that almost all species in the palatal odontogenic group have an intact domain of *Shh* expression (*Vonk et al., 2008*). Previous research indicated that in turtle embryos, the impairment of *Shh* signaling at an early stage arrests odontoblast development, thus contributing to the tooth loss of turtles (*Tokita, Chaeychomsri & Siruntawineti, 2013*). We speculate that the diversity of tooth patterns in turtles might affect the selective pressures of *Shh* in Testudines.

Desert hedgehog plays a role in spermatogenesis. It was previously found that *Dhh*-null male mice have small gonads, do not produce sperm, and their gender is reversed, as these males develop as phenotypic females (*Bitgood, Shen & McMahon, 1996*). In contrast, *Ihh* is one of the key genes associated with body development and is required for embryonic bone formation (*Shi et al., 2015*). Therefore, the different evolutionary rates of the *Dhh* and *Ihh* genes between limb and limbless clades may indicate different body development patterns. Moreover, the proteins encoded by *Shh*, *Dhh*, and *Ihh* are believed to have similar physiological effects, and that their differences during development may arise from varying patterns of expression (*Mcmahon, Ingham & Tabin, 2003*). Therefore,

the evolutionary differences in the three hedgehog genes among reptiles may be due to their different roles in the hedgehog signaling pathway.

## CONCLUSIONS

In summary, this study thoroughly investigated the structure and evolution of three hedgehog genes in reptiles for the first time. Characterization of these hedgehog members in reptiles indicates that the gene family was conserved in vertebrates. The selection pressure analysis of hedgehog genes indicated that three members had evolved, along with their vital function in body development. The data revealed that the three hedgehog genes underwent significant purifying selection. Interestingly, it is likely that the degeneration of snakes' limbs was a result of the regulation of multiple genes, rather than *Shh* alone. Additionally, *Dhh* and *Ihh* genes had faster evolutionary rates within different clades. Together, the findings of this study have provided extensive information regarding the molecular evolutionary history of the hedgehog gene family in reptiles and new insights for the study on reptilian adaptive evolution. Inferring gene function from public genome databases is not conclusive. Therefore, further functional experimentation is required to confirm our observations substantially.

## ACKNOWLEDGEMENTS

We would like to thank all those involved in the implementation of this study for their assistance, guidance, and support.

### Funding

This research was supported by the Special Fund for Forest Scientific Research in the Public Welfare (201404420) and the National Natural Science Fund of China (31872242, 31672313, 31372220). The funders had no role in study design, data collection and analysis, decision to publish, or preparation of the manuscript.

### Grant Disclosures

The following grant information was disclosed by the authors:
Special Fund for Forest Scientific Research in the Public Welfare: 201404420.
National Natural Science Fund of China: 31872242, 31672313 and 31372220.

### Competing Interests

The authors declare that they have no competing interests.

### Author Contributions

- Tian Xia performed the experiments, analyzed the data, contributed reagents/materials/analysis tools, authored or reviewed drafts of the paper, approved the final draft.
- Honghai Zhang conceived and designed the experiments, approved the final draft.
- Lei Zhang analyzed the data.
- Xiufeng Yang prepared figures and/or tables.

- Guolei Sun contributed reagents/materials/analysis tools.
- Jun Chen authored or reviewed drafts of the paper.
- Dajie Xu prepared figures and/or tables.
- Chao Zhao approved the final draft.

## Data Availability

The raw measurements are available in Table S2.

## Supplemental Information

Supplemental information for this article can be found online at http://dx.doi.org/10.7717/peerj.7613#supplemental-information.

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
