# Peer review of "Comparative and evolutionary analysis of the reptilian hedgehog gene family (Shh, Dhh, and Ihh)"

_PeerJ, doi:10.7717/peerj.7613_

## Round 0.1 · original submission · Major Revisions

Two reviewers felt the paper was very interesting but there is a need to improve the English writing and also to provide more details on the specific tests of natural selection used. Reviewer 1 feels that some additional tests may need to be conducted to address the issue of differences in selection pressure between reptile clades.

Reviewer 1 ·

Basic reporting

• The English needs significant improvement throughout. The overall structure of the manuscript is good and hypotheses somewhat clear, but could be better defined.
• The papers for the genomes used to obtain sequences for the study need to be cited.
• The figures are adequate, but formatting and readability could be improved.

Experimental design

• Overall research question is relevant and generally well defined; however, there are some methodological flaws.
• The exclusion of birds from the study is a serious flaw since birds are the sister group to crocodylians. There are, however, plenty of bird genomes that these sequences can be obtained from. I would recommend using a representative sampling of birds so as not to taxonomically bias the analysis by using all available bird genomes. There are also other reptile genomes that would be included to further increase the sample size, which will provide additional power to the selection tests.
• The authors seem to have two main hypotheses about the evolution of the hedgehog genes: that reptiles may be evolving differently than other vertebrates and that limbless species may also be under different selective pressure. I think these are reasonable hypotheses but they could be better tested. To test whether reptiles are evolved differently than other vertebrates the authors would need to include a representative sample of other vertebrates for comparison. The basis of the hypothesis could also be better explained. To test for an effect of limb loss it would be useful to include other limbless lineages besides snakes. There is at least one limbless lizard genome (Ophisaurus gracilis) that could be utilized for this purpose. Additionally, the PAML clade models would be better suited for these tests. See Baker et al Genetics 203:905–922 and Schott et al. MBE 2018 35:1376-1389 for examples. There are also additional HYPHY models that could be utilized as well, although the addition of the clade models would be sufficient. It is unclear why several of the lineage tests were performed and this should be explained.
• PAML sites model M8a should be compared to M7 to test for positive selective (not just the M8 vs M7 test, which only tests for addition of a site class)
• Datamonkey is a webserver that hosts various HYPHY programs. The names of the specific programs should be used (fine to indicate that they were used as implemented on the Datamonkey webserver)
• MEGA should not be used to estimate phylogenetic trees. Please use a more robust phylogenetic program (e.g., RaxML or PhyML for maximum likelihood)
• Reptiles did not arise from the amphibian ancestor. I believe the authors mean to say that the split between the two ancestors occurred roughly 330-310 mya

Validity of the findings

I expect the findings may change significantly after the methodological issues are resolved.

Additional comments

Overall I found this to be an interesting paper on the molecular evolution of the hedgehog genes in reptiles, a group that is severely understudied.

Reviewer 2 ·

Basic reporting

The Introduction rambles and is not coherent. It also includes many extraneous sentences that should be removed. Thus, the entire Introduction should be completely re-written so that it presents a coherent summary of the background on hedgehog genes in vertebrates and succinctly finishes with a sentence or two about what knowledge gap this paper attempts to fill.
Here are a couple specific examples of things to fix in the Introduction:
Lines 115-116: the authors should do more than merely state that “reptilian hedgehog signaling pathway has received little attention.” The authors should cite these early papers on this subject and review what their key findings were.
Lines 116-118: again, the authors should cite some key papers here and briefly mention their contributions, shortcomings, etc. This should represent a distinct paragraph that immediately precedes a paragraph describing what we know about reptilian hedgehog genes.

Experimental design

I am not an expert at detecting selection in genes but the bioinformatic methods employed by the authors appear appropriate to me.

Validity of the findings

I am not at all comfortable with the main Conclusion on lines 277-278: "In particularly, it is likely that the degeneration of the snake’s limbs is due to the regulation of multiple genes." I cannot find anything in their results that clearly shows this to be "likely."

Additional comments

This is a very interesting subject and the authors have a good bioinformatics-based approach to learn more about the evolution of vertebrate hedgehog genes. However, the manuscript, as written, is not yet suitable for publication in Peerj. The authors must try to improve the writing, especially the English, before it can be fairly evaluated by reviewers.

---

## Round 0.2 · Minor Revisions

The reviewers felt that the manuscript is greatly improved but they still feel some improvements are necessary. It will be important in the revision to enlist the help of an English writer so that you can improve the English. You should probably opt to pay for a professional service - it is not expensive and would greatly improve the writing and impact of your paper. Otherwise the reviewers have primarily pointed out some additional areas that could be presented more clearly, including some results that should be showcased more because of their general interest.

Reviewer 1 ·

Basic reporting

The writing is improved from the first version, but still needs additional improvement in my opinion. I've highlighted some of the particular statements below that should be addressed:

Ln 60-61: No positive selection site was identified by the PAML site model or the three methods in the Data Monkey Server collectively.

Ln 82: The three hedgehog genes have different functions in their expression patterns

Ln 83-85: Namely, the Shh gene plays a core role in the growth and patterning of the skeletal and nervous systems (Ingham & Mcmahon 2001a), and all are structures that have undergone remarkable morphological changes in primates, especially humans

Ln 99-100 : Previous research on the adaptive evolution of the hedgehog gene family in invertebrates indicated positive evolution,

Ln 153-154: Reptiles were distributed into 3 different orders of Reptilia

Ln 179-180 In order to determine whether hedgehog genes have faced identity or discrepant evolutionary pressure

Ln 187: Amino acids with selection pressure for site model M8 were identified using a

Ln 237: Comparatively, the diverse clades of the Shh gene were not reached the significant level

Ln 288-229 We set different orders as independent lineages, which showing diverse development characteristics.

Ln 267-268: Site models performed in PAML found that each codon of the Shh, Dhh, and Ihh sequences was under diverse selective pressures

Table 1 should be moved to the supplement.

Table 2: If I am interpreting correctly this is the branch model test and not a test of positive selection. Should show both the background and foreground w values so the difference can be compared. You say "clade of". Are you highlighting the entire clade (using $1) or just the branch leading to the clade (#1). Both are valid ways of testing for differences in selection, but this needs to be clarified. What is serpentes+OG? Is this the test including Ophisaurus? These comments apply to the presentation of the results from the other models as well.

Table 3 w values should be reported.

The issues with the tree topology appear to be due to the trees not being appropriately rooted.

Experimental design

My main concerns with the experimental design appear to have been largely addressed. However the explanation of the methods is still difficult to follow. I would recommend the authors use previous studies that employ similar methods as a template for how to explain the methods and present the results.

The clade model test seems to have been implemented using only a model that contains 4 different clade partitions. These should be redone with smaller numbers of partitions as well, ideally to test specific hypotheses.

Validity of the findings

Some of the conclusions made are still too strong and don't reflect the findings. These are largely unnecessary for the scope of the paper and should just be removed or highly modified. The title needs to be changed as genomic evidence of adaptive evolution is much too strong of a statement. I've highlighted some other problematic statements below:

Ln 69-71: Overall, this study has provided significative information regarding the evolution of the hedgehog gene family in reptiles and has provided new insight that will contribute to the protection and conservation of reptiles.

Ln 311: Together, the findings of this study have provided extensive information regarding the evolutionary history of the hedgehog gene family in reptiles and new insight for the conservation of reptilian populations.

Some potentially interesting results tend to be glossed over in favor of making very broad (and unnecessary) statements. Some of the specific results that I think deserve more attention are the apparent loss of genes in different groups, the tests for differences in selective patterns in limbless squamates, and the evidence for positive selection in turtles.

Additional comments

I appreciate that the authors have made a substantial effort to improve the manuscript and that the language barrier makes effectively communicating the results more difficult. I think with some additional editing and minor modifications to some of the analyses the manuscript will be suitable for publication.

Reviewer 2 ·

Basic reporting

I am sympathetic to the authors because I know that English is not their first language--it is very challenging to write a scientific paper in a foreign language. The authors have improved the English writing in this revision, but I still found many errors (words not spelled correctly or incorrectly used). I will provide some concrete examples as well as my other specific comments below:
line 59: what is "analogical topology"?
line 69: do you mean "significant information"?
line 78: this sentence needs to be re-written because it is misleading; it was not the genomes that were duplicated, but the hedgehog genes themselves- according to Kumar et al. they were duplicated two times: the first duplication event gave rise to desert hh and the second produced Indian hh and Sonic hh. The molecular mechanism for these tandem duplications was probably unequal crossing over.
lines 96-97: I don't think it is appropriate to suggest that these genes possibly promoted the successful diversification of vertebrates without providing some kind of clear rationale or citations.
line 98: do you mean "positive selection"? "positive evolution" does not make sense to me.
lines 112-113: I think you can remove this sentence about the divergence of reptiles from amphibians - I don't see the relevance to the current study.
lines 115-118: same here: I don't see the relevance of this statement to the paper so I suggest taking it out.
lines 162-165: this needs to be written more clearly because it is confusing at present. I understand that you used RAXML to construct a phylogenetic tree but the bootstraps are normally used to assess the statistical confidence of each node (or clade) in the tree.
lines 173-174: which previous studies? You need to cite them and provide more details on their key findings so that the reader can better understand the following sentence.
lines 176-178: I had trouble understanding this sentence - what is "identity or discrepant evolutionary pressure"? I suggest re-writing this sentence to improve its clarity.
lines 211-212: I think you mean to say "hedgehog gene tree"
lines 223-224: this sentence needs to be re-written because it is not clear.
line 234: this should be "Discussion"
line 253: I think you mean "species tree"
line 264: I think you mean "positively selected site"
line 278: I think you mean to say "because the mechanism of limb reduction in reptiles is..."
lines 299-301: I am not sure I understand this conclusion. I suggest re-writing this to improve the clarity.
I suggest increasing the size of the bootstrap values on the tree figures as they are too small to read.

Experimental design

The experimental design appears to be sound to me. I think the authors greatly improved their study by including all the additional genomes.

Validity of the findings

The findings appear to be ok to me.

Additional comments

This is a very interesting study and I commend the authors for their excellent work overall. I think if they further improve the writing to make everything more concise and clear, then their manuscript will be closer to being acceptable for publication in Peerj.

---

## Round 0.3 · accepted · Accept

Thank you for addressing the reviewers' comments and for improving the quality of the writing.